# Deletion of the Imprinted *Phlda2* Gene Increases Placental Passive Permeability in the Mouse

**DOI:** 10.3390/genes12050639

**Published:** 2021-04-25

**Authors:** Emily Angiolini, Ionel Sandovici, Philip M. Coan, Graham J. Burton, Colin P. Sibley, Abigail L. Fowden, Miguel Constância

**Affiliations:** 1University of Cambridge Metabolic Research Laboratories and MRC Metabolic Diseases Unit, Institute of Metabolic Science, Addenbrookes Hospital, Cambridge CB2 0QQ, UK; emily.angiolini@earlham.ac.uk (E.A.); is299@cam.ac.uk (I.S.); 2Department of Obstetrics and Gynaecology, National Institute for Health Research Cambridge Biomedical Research Centre, Cambridge CB2 0SW, UK; 3Epigenetics Programme, The Babraham Institute, Babraham Research Campus, Cambridge CB22 3AT, UK; 4Centre for Trophoblast Research, Department of Physiology, Development and Neuroscience, University of Cambridge, Cambridge CB2 3EG, UK; p.m.coan.02@cantab.net (P.M.C.); gjb2@cam.ac.uk (G.J.B.); alf1000@cam.ac.uk (A.L.F.); 5Maternal and Fetal Health Research Centre, Division of Developmental Biology and Medicine, School of Medical Sciences, Faculty of Biology, Medicine and Health, The University of Manchester, Manchester M13 9WL, UK; colin.sibley@manchester.ac.uk; 6Manchester Academic Health Science Centre, St. Mary’s Hospital, Manchester University NHS Foundation Trust, Manchester M13 9WL, UK; 7Department of Physiology, Development and Neuroscience, University of Cambridge, Cambridge CB2 3EG, UK

**Keywords:** *Phlda2*, genomic imprinting, placenta, fetal growth, passive permeability

## Abstract

Genomic imprinting, an epigenetic phenomenon that causes the expression of a small set of genes in a parent-of-origin-specific manner, is thought to have co-evolved with placentation. Many imprinted genes are expressed in the placenta, where they play diverse roles related to development and nutrient supply function. However, only a small number of imprinted genes have been functionally tested for a role in nutrient transfer capacity in relation to the structural characteristics of the exchange labyrinthine zone. Here, we examine the transfer capacity in a mouse model deficient for the maternally expressed *Phlda2* gene, which results in placental overgrowth and a transient reduction in fetal growth. Using stereology, we show that the morphology of the labyrinthine zone in *Phlda2*^−/+^ mutants is normal at E16 and E19. In vivo placental transfer of radiolabeled solutes ^14^C-methyl-D-glucose and ^14^C-MeAIB remains unaffected at both gestational time points. However, placental passive permeability, as measured using two inert hydrophilic solutes (^14^C-mannitol; ^14^C-inulin), is significantly higher in mutants. Importantly, this increase in passive permeability is associated with fetal catch-up growth. Our findings uncover a key role played by the imprinted *Phlda2* gene in modifying placental passive permeability that may be important for determining fetal growth.

## 1. Introduction

Genomic imprinting is an epigenetic phenomenon that leads to allele-specific expression depending on the parent-of-origin of the allele [1]. Genomic imprinting is thought to have evolved independently in flowering plants with endosperm and in placental mammals, both of which have specialist structures to support the development of the offspring [2]. In mammals, there is an evolutionary link between genomic imprinting and placentation, strongly supporting the hypothesis that imprinted genes play key roles in allocation of maternal resources via the placenta. Reinforcing the notion that imprinted genes have a special role in placental biology, a large proportion of the approximately 100 imprinted genes identified to date in mice and humans are highly expressed in the placenta [3]. Additionally, mouse models of loss-of-imprinting, leading to either reactivation of the silent allele or silencing of the active allele, have uncovered a number of placental phenotypes, ranging from abnormal growth and structure to changes in placental transfer capacity and endocrine function (reviewed by [4,5,6]).

*Phlda2* (pleckstrin homology-like domain family A member 2, also known as *Ipl*—imprinted in placenta and liver) is a maternally expressed gene located in a cluster of imprinted genes on distal chromosome 7 in the mouse, with shared synteny to the human chromosome 11p15.5 region. During early mouse development, prior to gastrulation, *Phlda2* mRNA expression is restricted to the extra-embryonic ectoderm and the ectoplacental cone [7]. In the definitive placenta (from E10.5), expression of PHLDA2 at the mRNA and protein level persists only in the labyrinthine trophoblast until mid-gestation. It then declines markedly after E14.5, except in the chorionic plate [8,9]. *Phlda2* expression is also observed at high levels in the visceral yolk sac endoderm until mid-gestation. In fetal tissues, *Phlda2* mRNA levels are moderate in liver and considerably lower in kidney, lung and limb [8]. In human placentae, PHLDA*2* protein is highly expressed in the villous cytotrophoblast [10]. By contrast, low levels are present in the cytotrophoblast cells columns of anchoring villi during early pregnancy and in the extravillous trophoblast derived from them [9,10]. PHLDA2 protein consists of a single pleckstrin homology (PH) domain with short N- and C-terminal extensions, which demonstrates binding to phosphatidylinositol phosphate (PIP) in vitro and is therefore likely involved in intracellular trafficking or signaling [11]. By homology to mouse PHLDA1/TDAG51, PHLDA2 may play a role in FAS-mediated apoptosis, which is consistent with a potential growth-inhibitory effect [12].

In mice, maternal deletion of the *Phlda2* gene (*Phlda2*^−/+^) leads to placentomegaly, with expansion of the junctional zone, independent of the genetic background [9,13,14]. Additionally, *Phlda2*^−/+^ leads to a transient reduction in fetal weight in late gestation on the C57BL/6J genetic background [9,14] and to a global fetal growth restriction on the 129S2/SvHsd genetic background affecting both the wild-type and the mutant fetuses [13]. Within the junctional zone, *Phlda2*^−/+^ mutants have more glycogen stored and the expression of several placental hormone genes is increased [13]. Increased *Phlda2* expression, achieved by loss-of-imprinting or insertion of a single-copy transgene, results in placental growth restriction, with loss of the spongiotrophoblast lineage and reduced placental glycogen accumulation [14,15]. Additionally, elevated *Phlda2* expression leads to asymmetric fetal growth restriction [15,16]. In humans, the expression levels of *PHLDA2* are negatively correlated with birthweight [17], and are elevated in cases of fetal growth restriction (FGR) [18,19,20]. Furthermore, maternal inheritance of a 15 bp variant at the *PHLDA2* promoter region that reduces its activity leads to increased birth weight [21], further demonstrating the importance of the finely balanced expression of this imprinted gene.

The overall aim of this study was to investigate the contribution of the placenta to the transitory reduction in fetal growth observed in *Phlda2*^−/+^ mutants [9]. To achieve this goal, we characterized the morphological and transfer properties of *Phlda2*^−/+^ placentae at two stages of late gestation (E16 and E19). Specifically, we used stereological techniques to establish the effects of *Phlda2*^−/+^ mutation on placental labyrinthine zone morphology and to estimate placental diffusion capacity, as well as placental transfer assays to determine placental transfer capacity related to passive permeability to hydrophilic solutes, facilitated diffusion and active transport in vivo.

## 2. Materials and Methods

### 2.1. Mice

The *Ipl*^loxP^ transgenic lines were produced elsewhere [9] and maintained in a C57BL/6J background. In all experiments, the mutant alleles were transmitted by a heterozygous mother carrying the deleted allele on her paternal chromosome, crossed with C57BL/6J wild-type males, giving the genotypes −/+ (subsequently called *Phlda2*^−/+^) and +/+ (subsequently called *Phlda2*^+/+^). Growth kinetics studies, placental transfer assays and stereological analyses were performed in −/+ versus +/+ littermates. Mice were fed a standard chow diet and housed with a 12 h light/dark cycle in a temperature-controlled room (22 °C). For timed matings, the day of detection of a vaginal plug was noted as embryonic day 1 (E1). All studies were performed at the embryonic days E16 and E19.

### 2.2. Genotyping

All genotyping was performed by PCR with DNA extracted from yolk sacks, using the Expand High Fidelity PCR system (Roche) and standard cycling conditions according to the manufacturer’s recommendations. *Phlda2*^−/+^ offspring were identified using primers: Phlda2F: 5′-TCAAGCAATGGGTAAGGG-3′ and Phlda2R: 5′-TCCATCATTAGAGACAGCCA-3′ to amplify a 362 bp fragment across the deletion, with an additional forward primer (5′-GTTATTCCCTCTCCACATCC-3′) used to amplify a 639 bp product from the wild-type alleles (annealing temperature is 55 °C).

### 2.3. Placental Stereology

Stereology was performed on placentae collected at E16 and E19 and undertaken blinded to genotype. Placentae were hemisected using a double-edged razor blade, each half weighed, and immediately fixed. The stereology analysis was performed according to the protocol described previously [22]. Briefly, half of each placenta was fixed in 4% paraformaldehyde and embedded in paraffin wax, sectioned and stained using a standard hematoxylin and eosin protocol. The corresponding placental halves were fixed for 6 h with 4% glutaraldehyde and embedded in resin, sectioned and stained with methylene blue. The paraffin and resin-embedded counterparts were then analyzed using the Computer Assisted Stereology Toolbox (CAST) 2.0 system from Olympus (Ballerup, Denmark). The paraffin sections allow measurements for absolute placental volumes and volumes of the three main placental zones (labyrinthine zone—Lz, junctional zone—Jz and decidua basalis—Db) using the Cavalieri principle, as well as Jz/Lz volume ratios. Resin sections allow for detailed measurements of the labyrinthine zone: volume of maternal blood spaces (MBS), fetal capillaries (FC) and labyrinthine trophoblast (LT), the surface areas of the MBS and FC and the thickness of the labyrinthine interhemal membrane (IMT), as well as calculating the theoretical diffusion capacity (TDC) and the specific diffusion capacity (SDC), as described [22]. TDC indicates the potential of the labyrinthine interhemal membrane for passive diffusion, and SDC is a measure of how effective the interhemal membrane of the placenta is at meeting fetal needs [22].

### 2.4. Placental Transfer Assays of Radio-Labelled Solutes

Placental transfer assays (PTAs) were performed in pregnant heterozygous females crossed with C57BL/6J males at E16 or E19 using the following radiolabeled tracers: ^14^C-mannitol, ^14^C-inulin-carboxyl and the non-metabolizable analogues, ^14^C-methyl-D-glucose and ^14^C-methyl-aminoisobutyric acid (MeAIB; a specific substrate of System A amino acid transporters). Details regarding PTAs have been described elsewhere [23,24]. Radioactive counts in each fetus (measured as DPM—disintegrations per minute) were used to calculate the amount of radioisotope transferred per gram of Lz or per gram of fetus. The weight of Lz was estimated by multiplying individual placental weights with an average Lz percentage value determined by placental stereology for each genotype and gestational stage (from 3 litters, as previous described for similar conversions [25]). Average values for *Phlda2*^+/+^ controls and *Phlda2*^−/+^ mutant fetuses within a litter were calculated and expressed as the *Phlda2*^−/+^/*Phlda2*^+/+^ ratio for that litter. These values could be used to calculate a mean for all litters at E16 and E19. The fetal accumulation of radioisotope expressed relative to Lz weight and plotted as the *Phlda2*^−/+^/*Phlda2*^+/+^ ratio gives a relative measure of Lz transfer of the solute. The fetal accumulation of radioisotope expressed relative to fetal weight gives a relative measure of the amount of solute received by the fetus.

### 2.5. Statistical Analysis

No explicit power analyses were used to predetermine sample size. All normally distributed data were analyzed by means of two-way analyses of variance with “litters” and “genotype” as the two factors. Data are expressed as means ± standard error of the mean (SE). For data representing radioactive counts, a logarithmic transformation was carried out before statistical analysis. The summary data from these experiments were then presented as ratios, together with 95% confidence limits or as DPM/g Lz or DPM/g fetus ± SE.

## 3. Results

### 3.1. Phlda2^−/+^ Mutants Show Placental Overgrowth Associated with Transitory Reduction in Fetal Growth

Maternal inheritance of *Phlda2*^−/+^ deletion on the C57BL/6J genetic background led to placental overgrowth and a transient reduction in fetal growth as previously reported [9,14]. Mutant placentae were on average ~129% of *Phlda2*^+/+^ littermates at both E16 and E19 of gestation (Table 1) (*p* < 0.001), whereas there was a modest but significant reduction in fetal weight to 96% at E16 (*p* < 0.05), which was lost by E19 (99%, *p* = NS, not significant). As a result, the *Phlda2*^−/+^ mutants displayed decreased fetal-to-placental (F/P) weight ratios in comparison to *Phlda2*^+/+^ control littermates (Table 1).

### 3.2. Placental Overgrowth in Phlda2^−/+^ Mutants Does Not Affect the Labyrinthine Zone

Next, we investigated whether the placental overgrowth, which was previously reported to be restricted to the endocrine zone (or junctional zone), also affected the transport zone (labyrinthine zone), by performing in-depth stereological analysis throughout the whole of the placenta. The marginal increases in Lz volume (110% and 115% of *Phlda2*^+/+^ littermates at E16 and E19, respectively) were not statistically significant (Table 2). This finding is in contrast to the significant increases in absolute volume of the Jz to 222% and 155% of *Phlda2*^+/+^ control littermates at E16 and E19, respectively (*p* < 0.01). As a result, the Jz/Lz ratio increased almost two-fold in *Phlda2^−/+^* at E16 (*p* < 0.01), but was similar between the two genotypes at E19 (129%; *p* = NS) (Table 2). The latter finding is mainly due to the well-established changes in the proportions of the two zones near term (i.e., a decrease in the junctional zone and an increase in the labyrinthine zone, as described previously [22]).

Within the Lz, further stereological analyses were made on resin sections, namely absolute measurements of the transport epithelium: labyrinthine trophoblast (LT), maternal blood spaces (MBS) and fetal capillaries (FC). Of these three Lz components (LT, MBS and FC), only the volume of MBS was significantly increased in *Phlda2^−/+^* placentae, specifically at E19 (151%; *p* < 0.05) (Table 3). However, the surface areas occupied by MBS, LT, FC were normal, as was the interhemal membrane thickness (IMT) at both E16 and E19 (Table 3). The theoretical diffusion capacity (TDC) and the specific diffusion capacity (SDC) of the *Phlda2^−/+^* placentae, which depend to a great extent on the IMT, were not altered significantly at either E16 or E19 (Table 3). Thus, the *Phlda2^−/+^* placentae have an enlarged Jz, while the structure and size of the Lz compartment is normal, at least at the level of resolution provided by light microscopy-based stereology.

### 3.3. Phlda2^−/+^ Placentae Show Increased Passive Permeability to Hydrophilic Solutes

To investigate the exchange properties of the *Phlda2^−/+^* placentae, we started by assessing the diffusional characteristics using the inert tracers ^14^C-mannitol and ^14^C-inulin, which can only cross the placenta by passive diffusion. Surprisingly, we found that the overgrown *Phlda2*^−/+^ placenta had a pronounced increase in its passive permeability relative to *Phlda2*^+/+^ controls, despite unaltered surface area of the Lz components and normal IMT, TDC and SDC compared to wild-types. Accordingly, the amounts of ^14^C-mannitol and ^14^C-inulin transferred per gram Lz were significantly increased to 154% and 156% of the *Phlda2*^+/+^ control values at E16, respectively (Figure 1A and Appendix A). Additionally, *Phlda2*^−/+^ placentae transferred increased amounts of the larger inert tracer ^14^C-inulin (134%; *p* < 0.01), but not ^14^C-mannitol, at E19 (Figure 1A and Appendix A). The accumulation of the two passive diffusion markers per gram of fetus was significantly increased at both E16 and E19, i.e., *Phlda2^−/+^* fetuses received an excess of the radioactive labels for their size relative to *Phlda2*^+/+^ controls (Figure 1A and Appendix A). These results strongly suggest that the *Phlda2^−/+^* placentae are more permeable to hydrophilic solutes than *Phlda2*^+/+^ littermates at both gestational ages.

Next, we investigated if other major transport processes, i.e., active transport and facilitated diffusion, were also affected in the *Phlda2*^−/+^ mutants, by assessing the in vivo flux of representative tracers, ^14^C-MeAIB and ^14^C-methyl-D-glucose, respectively. We found that the transfer of ^14^C-methyl-D-glucose per gram of Lz was similar between *Phlda2*^−/+^ and *Phlda2*^+/+^ at both E16 and E19 (Figure 1B and Appendix A). Accumulation of ^14^C-methyl-D-glucose per gram of fetus was also similar between the two genotypes at both gestational ages, i.e., mutant fetuses received an appropriate amount of ^14^C-methyl-D-glucose for their size (Figure 1B and Appendix A). Additionally, ^14^C-MeAIB transfer per gram of Lz placenta and ^14^C-MeAIB accumulation per gram of fetus were normal at both E16 and E19 (Figure 1B and Appendix A).

We conclude that maternal transmission of the *Phlda2*^−/+^ deletion leads to placentae that are more permeable to hydrophilic solutes, leading to altered diffusional flux of such solutes to and from the growing fetus. This alteration seems specific to passive permeability, as transplacental flux by facilitated diffusion and active transport is normal at both E16 and E19.

## 4. Discussion

Maternal–fetal exchange by passive diffusion is responsible for a significant proportion of the total solute flux across the hemochorial placenta and is therefore thought to be a significant determinant of fetal growth [26,27]. Indeed, passive diffusion is responsible for the transfer of water, ions, minerals and water-soluble vitamins needed for enzyme activities [26]. This study uncovered a novel role for the imprinted *Phlda2* gene as a putative negative regulator of the diffusional permeability characteristics of the mouse placenta. Our findings are based on measurements of the maternal–fetal transfer of two inert hydrophilic molecules of increasing size (D_w_ values of 9.9 cm^2^/s × 10^6^ and 2.6 cm^2^/s × 10^6^ for ^14^C-mannitol and ^14^C-inulin, respectively, see [23]) in *Phlda2*-deficient mice, which showed significantly enhanced transfer during late gestation. Our study adds *Phlda2* to the growing list of imprinted genes with roles in the developmental regulation of diffusional exchange characteristics of the placenta. To our knowledge, this study is the first to report the action of an imprinted gene in determining maternal–fetal exchange of solutes solely by effects on passive diffusion.

In this study, we set out to explore aspects related to the placental phenotype of the *Phlda2*^−/+^ mouse model. First, the expression of *Phlda2* is observed primarily in the trophoblast cells within the Lz, but the placental phenotypes, including placental overgrowth, were previously reported to be restricted to the *Phlda2*-non-expressing Jz. One of the objectives of our study was, therefore, to investigate if the lack of normal *Phlda2* expression in the Lz had any functional impact in this layer, by performing in-depth stereological analyses focused on the morphological characteristics of the Lz that determine transfer capacity. Second, *Phlda2*^−/+^ mutation results in a modest and transitory reduction in fetal growth. We wanted to establish if, and how, the placenta contributes to the fetal growth phenotype, primarily by analyzing the trans-placental flux of nutrients and other solutes.

Consistent with previous findings in C57BL/6J genetic background [9], we found that *Phlda2*^−/+^ conceptuses show overgrowth of the placenta by ~30% at both E16 and E19. In our study, however, fetuses were approximately 4% lighter at E16 and of normal weight by E19, whereas Frank et al. reported a higher degree of FGR, with fetuses ~13% lighter at E17, with no data available at E19 or birth [9]. Based on our study alone, these observations suggest that mutant *Phlda2*^−/+^ fetuses undergo an initial phase of fetal growth retardation, followed by intra-uterine catch-up growth. Interestingly, on the 129S2/SvHsd genetic background, *Phlda2*^−/+^ fetuses did not show any weight difference compared to their *Phlda2*^+/+^ littermate controls at E14.5, E16.5 and E18.5 [13]. Additionally, placental overgrowth in *Phlda2*^−/+^ mutants was more pronounced on the 129S2/SvHsd genetic background compared to the C57BL/6J genetic background, with a maximal increase of 50% at E16.5 [13]. However, it was also observed that at E18.5 both *Phlda2*^+/+^ and *Phlda2*^−/+^ fetuses were about 15% lighter when compared to wild-type litters on the 129S2/SvHsd genetic background, demonstrating that *Phlda2*^−/+^ mutation is affecting the phenotype in wild-type conceptuses of the same litter [13]. However, in our study, *Phlda2*^+/+^ fetuses from mixed litters were of similar weight to those from wild-type litters on the C57BL/6J genetic background collected subsequently, at both E16 (0.398 ± 0.005 (n = 48) versus 0.396 ± 0.003 (n = 70), *p* = 0.91) and E19 (1.079 ± 0.009 (n = 88) versus 1.108 ± 0.011 (n = 37), *p* = 0.15). Our stereological analyses of the placental phenotype confirmed that deletion of maternal *Phlda2* allele leads to a disproportionate overgrowth of placental Jz [9,13]. However, we also provide evidence that the structure of Lz was remarkably normal, with the notable exception of maternal blood spaces that occupy a larger volume at E19 in mutants. The finding that the structure of Lz was largely normal was surprising, given that PHLDA2 expression is restricted to type III (fetal facing) and type II (proliferating) trophoblast cells of Lz [9]. These findings taken together are consistent with the suggestion that the overgrowth of Jz is controlled by signaling events taking place in Lz by unknown mechanisms [9]. An alternative explanation is that *Phlda2* limits the expansion of progenitors contributing to the Jz, as previously proposed [15].

As described in detail elsewhere [23,28], the rate of transfer across the placenta of an inert uncharged solute is described by Fick’s Law of diffusion and gives a measure of the passive permeability. Thus, flux is determined directly by the surface area for exchange and the maternal–fetal concentration gradient, and inversely by the thickness of the barrier and the size of the molecule (usually determined by its diffusion coefficient in water at 37 °C). Under the conditions of our experiments, where transfer was measured as a unidirectional flux, the only variables expected to alter the transfer of the ^14^C-mannitol and ^14^C-inulin would be the surface area available for exchange and/or the thickness of the exchange barrier. Surprisingly, therefore, we measured an increase in the diffusional permeability of these solutes in the absence of any change in the stereologically measured area or thickness of the IMT, as well as TDC and SDC. However, the previously reported dependence of diffusional transfer on the size of such hydrophilic solutes [23,27], which cannot easily cross the hydrophobic plasma membrane, suggests that there must be an extracellular water-filled route across the syncytiotrophoblast through which diffusion takes place. The nature of this extracellular water-filled route is uncertain and its dimensions would likely differ from that of the stereologically measured barrier surface area and thickness. In humans, transtrophoblastic channels [29] and syncytial denudations [30] have been suggested as providing such a route, but there is no study, of which we are aware, which addresses this issue in the mouse. We speculate that the increase in ^14^C-mannitol and ^14^C-inulin diffusion observed here in the *Phlda2*^−/+^ mutant, in the absence of a change in total Lz surface area or thickness, is related to a change in the length or radius of such putative routes in the trophoblast layers. We further speculate that ^14^C-inulin transfer remaining higher in the mutant at E19, whereas that of ^14^C-mannitol is normal at this gestation, is related to the number of a population of narrower channels, through which most of the diffusion of the smaller ^14^C-mannitol molecule takes place, increasing at E16 and returning to normal at E19, whereas the number of a population of wider channels, through which most of the ^14^C-inulin diffusion takes place, is increased at E16 and remains so at E19. Heterogenous channel populations in the hemochorial placenta have previously been suggested [31]. As discussed in [28], the nature of the diffusional route for hydrophilic solutes across the placenta is a fundamental piece of information lacking in our understanding of placental exchange physiology; the *Phlda2*^−/+^ mutant might provide a useful tool for dissecting the nature and regulation of this route.

It is tempting to speculate that the increased passive permeability of *Phlda2*^−/+^ placentae contributes to the intra-uterine catch-up fetal growth that is observed prior to E19. Although placental weight reaches a plateau after E16, or even decreases slightly, placental Lz, including the interchange surface area, continues to expand until term [22]. Placental passive permeability also increases in absolute terms between E16 and E19 (Appendix A and [23]). *Phlda2^−/+^* fetuses received an excess of the radioactive labels for their size relative to *Phlda2*^+/+^ controls, strongly suggesting that there is increased flux of nutrients reaching the fetus. Overall, according to Fick’s Law, the effect of an increased passive permeability on the transfer of individual solutes will be dependent on maternal/fetal electrochemical gradients, about which we know little in the mouse. However, to illustrate the point, if we assume the maternal plasma glucose concentration is higher than in fetal plasma, as it is in most species, then increased permeability will increase glucose flux to the fetus, in addition to that transferred by facilitated diffusion. However, for amino acids where the concentration gradient is usually higher in the fetal plasma than in the maternal plasma (due to active transport), then increased permeability could actually decrease net amino acid flux to the fetus, due to increased diffusional back flux in the fetal to maternal direction.

Our study is inconclusive regarding the contribution of the placenta to the reduced fetal weight observed at E16 in our study and at E17 in [9]. From our stereological studies and placental transfer studies, we conclude that the transitory reduction in fetal growth is not caused by reduced transport capacity to the fetus. However, we cannot rule out other explanations such as increased partitioning of maternal resources to the placenta to sustain the expansion of the Jz layer, as previously proposed [13]. Indeed, the placenta itself consumes approximately 35–40% of the oxygen and glucose removed from the maternal uterine circulation by the feto-placental unit [32,33,34]. Placental metabolism determines the types and quantities of nutrients transferred to the fetus [35]. Placental metabolism depends on the size of placenta and responds to a number of environmental cues, including diet, hormones and oxygen [36,37,38,39,40]. An increase in placental size overall would put a greater strain on the maternal nutrient resources, which is limited and known as maternal constraint [41,42]. Several tissues, especially the liver and kidney, express significant levels of *Phlda2* mRNA [8]. In part, the changes in fetal growth may also relate to altered *Phlda2* expression in the fetal tissues, so conditional deletions of fetal and placental *Phlada2* would be needed to differentiate between fetal and placental contribution to the changes in the fetal growth trajectory. These could be achieved using, for example, the *Cyp19*-Cre line that is active in the early diploid trophoblast cells from which both placental Lz and Jz derive [43].

We have previously shown that the developmental regulation of diffusional exchange characteristics of the placenta is an important mechanism by which imprinted genes influence placental nutrient supply and fetal growth (summarized in Table 4). Deletion in the mouse of the paternally expressed placental-specific *Igf2* P0 transcript (*Igf2*P0^+/−^) [44] led to a small-sized placenta and reduced placental passive permeability for three inert hydrophilic solutes of increasing sizes (^14^C-mannitol, ^51^CrEDTA, and ^14^C-inulin) at E19 of gestation, which was associated with an increase in thickness of the exchange barrier compared to littermate controls [23] (Table 4). When all *Igf2* transcripts were deleted throughout the conceptus (*Igf2*null^+/−^), the increase in the thickness of the barrier was milder compared to *Igf2*P0^+/−^ mutants [45] and only the permeability for the largest tracer (^14^C-inulin) was significantly reduced at E19 of gestation [45] (Table 4). Interestingly, in both *Igf2* models (*Igf2*P0^+/−^ and *Igf2*null^+/−^) we observed alterations in the transplacental flux of ^14^C-methyl-D-glucose and/or ^14^C-MeAIB, as well as placental expression of amino acid and glucose transporters [24]. These findings strongly suggest that the small mutant *Igf2*P0^+/−^and *Igf2*null^+/−^ placentae are capable of functional compensatory adaptations in response to either increased or decreased fetal demands for growth, respectively [24]. Overexpression of *Igf2*, achieved through a 13 kb deletion that included the *H19* gene and the *H19*/*Igf2*-imprinting control region ICR1 (*H19*^Δ13^), led to an enlarged placenta, with pronounced reductions in placental nutrient transfer by the three mechanisms, including passive permeability, with the amounts of ^14^C-mannitol and ^14^C-inulin transferred by the placenta significantly reduced in mutants at both E16 and E19 [42] (Table 4). In this model of *Igf2* overexpression, the thickness of the exchange barrier in mutants was not different from control littermates. *Rtl1* (retrotransposon-like 1) is a paternally expressed gene that has a demonstrated role in altering placental passive permeability [46]. Accordingly, paternal deletion of *Rtl1* led to a significant reduction in placental passive permeability for ^14^C-inulin at E16, likely explained by areas of placental infarction observed in these mutants [46]. Altogether, these findings show that imprinted genes modify placental passive permeability, through several possible mechanisms, from regulation of the thickness of exchange barrier (*Igf2*P0) and altered number or size of pores (*H19*/*Igf2*, and *Phlda2* as first demonstrated here), to developmental defects (*Rtl1*). To our knowledge, *Phlda2* is the only imprinted gene described so far with a role in modulating maternal-fetal exchange of solutes solely by effects on passive permeability. We propose that an important novel function for this maternally expressed gene is to limit the allocation of maternal resources to the growing fetus by reducing the permeability characteristics of the placenta. This ‘strategy’ adopted by *Phlda2* is consistent with the general roles played by maternally expressed genes in the offspring as proposed by the two most widely accepted evolutionary theories of imprinting, i.e., the kinship and maternal-offspring coadaptation.

We also note that all imprinted genes that were assessed to date for placental passive permeability properties in transgenic studies show evidence for impaired exchange of diffusional characteristics (*Rtl1*, *H19*/*Igf2*, *Igf2*P0 and now *Phlda2*—see Table 4). At present, there is no evidence of interaction of imprinted genes in the regulation of passive permeability. However, it has not escaped our attention that the imprinted gene network (IGN), regulated by the zinc-finger transcription factor PLAGL1 [47], includes IC1 and IC2 genes (e.g., *Igf2*, *H19*) but also *Rtl1*, all of which have a major impact on placenta passive permeability. Moreover, *H19* is a transregulator of the IGN, with the knock-out of *H19* causing expression changes in a number of IGN genes (including *Rtl1*, *Igf2* and *Cdkn1c*) [47]. It will be important to define, in future studies, how many other imprinted genes may play roles in passive diffusion and perform genetic studies targeted to the placenta to provide evidence for direct or indirect interactions between IGN genes in the regulation of passive diffusion. At the moment, we propose that *Phlda2* may play a unique and specific role in the control of pore size/number, but methods to visualize and measure such pores are urgently needed.

## 5. Conclusions

Overall, in this study, we filled an important gap in our knowledge about the role played by the imprinted *Phlda2* gene in the placenta. Our results reiterate the important roles played by the imprinted genes in controlling the exchange characteristics of the mouse placenta and the complex relationship between these functional properties and the detailed morphological structure of the interchange layer of placenta. By influencing placental passive permeability, *Phlda2* may play an important role in fine-tuning fetal growth to reach its genetically determined potential. Further studies are needed to define the specific molecular mechanisms by which the *Phlda2* gene acts on placental passive permeability.

## Figures and Tables

**Figure 1 genes-12-00639-f001:**
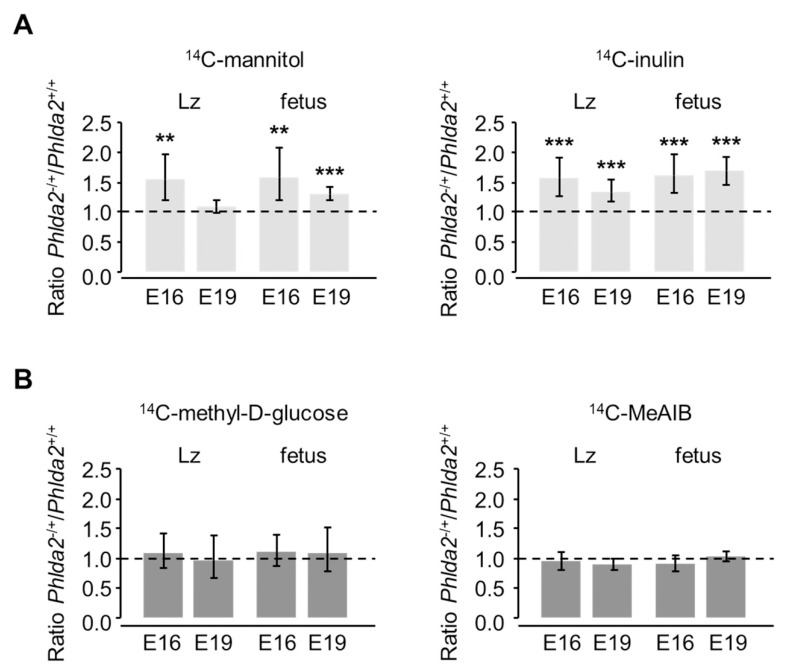
Placental transfer of (**A**) passive permeability markers (^14^C-mannitol and ^14^C-inulin) and (**B**) facilitated and active transport markers (^14^C-methyl-D-glucose and ^14^C-MeAIB, respectively) in *Phlda2*^−/+^, calculated as a ratio of *Phlda2*^−/+^ to *Phlda2*^+/+^ transfer, expressed either per gram of labyrinthine zone (Lz) or per gram of fetus at two gestational ages (E16 and E19). Ratios >1 indicate increased transfer by the *Phlda2*^−/+^ mutant placenta with respect to either Lz or fetal weight (E16: ^14^C-mannitol, n = 12 *Phlda2*^+/+^ and n = 15 *Phlda2*^−/+^ from 4 litters; ^14^C-inulin, n = 31 *Phlda2*^+/+^ and n = 32 *Phlda2*^−/+^ from 8 litters; ^14^C-methyl-D-glucose, n = 20 *Phlda2*^+/+^ and n = 17 *Phlda2*^−/+^ from 6 litters; ^14^C-MeAIB, n = 16 *Phlda2*^+/+^ and n = 17 *Phlda2*^−/+^ from 5 litters. E19: ^14^C-mannitol, n = 26 *Phlda2*^+/+^ and n = 35 *Phlda2*^−/+^ from 9 litters; ^14^C-inulin, n = 22 *Phlda2*^+/+^ and n = 18 *Phlda2*^−/+^ from 5 L; ^14^C-methyl-D-glucose, n = 18 *Phlda2*^+/+^ and n = 20 *Phlda2*^−/+^ from 5 litters; ^14^C-MeAIB, n = 48 *Phlda2*^+/+^ and n = 42 *Phlda2*^−/+^ from 13 litters. Bars indicate 95% confidence limits. ** *p* < 0.01; *** *p* < 0.001.

**Table 1 genes-12-00639-t001:** Fetal weights, placental weights and F/P weight ratios for *Phlda2*^−/+^ mutants and *Phlda2*^+/+^ controls.

Gestational Age	Genotype	N	Fetal Wet Weight (g) ± SE	Placental Wet Weight (g) ± SE	F/P Weight Ratios ± SE
E16	*Phlda2* ^+/+^	48	0.398 ± 0.005	0.100 ± 0.002	4.007 ± 0.083
	*Phlda2* ^−/+^	49	0.384 ± 0.005	0.129 ± 0.002	3.099 ± 0.081
	(% of *Phlda2*^+/+^)		96.4 *	129.7 ***	77.3 ***
E19	*Phlda2* ^+/+^	88	1.079 ± 0.009	0.082 ± 0.001	13.395 ± 0.227
	*Phlda2* ^−/+^	94	1.069 ± 0.009	0.106 ± 0.001	10.247 ± 0.172
	(% of *Phlda2*^+/+^)		99.1	129.3 ***	76.5 ***

F/P weight ratios = Fetal wet weight/Placental wet weight. * *p* < 0.05; *** *p* < 0.001.

**Table 2 genes-12-00639-t002:** Absolute volumes of components in the placenta of *Phlda2*^−/+^ mutants and *Phlda2*^+/+^ controls at two gestational stages (E16 and E19).

	E16			E19		
	*Phlda2* ^+/+^	*Phlda2* ^−/+^	% of *Phlda2*^+/+^	*Phlda2* ^+/+^	*Phlda2* ^−/+^	% of *Phlda2*^+/+^
Placenta	93.49 ± 3.215	139.36 ± 5.754	149 **	88.73 ± 7.377	116.41 ± 1.677	131 *
Lz	40.71 ± 2.151	44.89 ± 1.575	110	44.13 ± 5.668	50.60 ± 2.122	115
Jz	32.14 ± 0.172	71.31 ± 8.042	222 **	30.54 ± 1.176	47.26 ± 2.597	155 **
Db	18.13 ± 1.499	19.78 ± 4.538	109	11.07 ± 1.603	13.82 ± 1.535	125
Jz/Lz	0.80 ± 0.049	1.58 ± 0.138	198 **	0.73 ± 0.110	0.94 ± 0.081	129

Values are mean ± SE volume (mm^3^); n = 9 per genotype at E16 and n = 6 per genotype at E19 from 3 litters/group; Lz—labyrinthine zone; Jz—junctional zone; Db—decidua basalis; * *p* < 0.05; ** *p* < 0.01.

**Table 3 genes-12-00639-t003:** Absolute quantities of Lz components in *Phlda2*^−/+^ and *Phlda2*^−/+^ placentae at E16 and E19.

	E16			E19		
	*Phlda2* ^+/+^	*Phlda2* ^−/+^	% of *Phlda2*^+/+^	*Phlda2* ^+/+^	*Phlda2* ^−/+^	% of *Phlda2*^+/+^
LT	26.07 ± 0.139	29.57 ± 1.418	113	28.92 ± 5.013	30.80 ± 1.059	107
MBS	7.37 ± 1.291	7.86 ± 1.156	107	7.47 ± 0.754	11.27 ± 0.938	151 *
FC	7.27 ± 1.384	7.45 ± 0.154	103	7.73 ± 1.565	8.52 ±1.469	110
MBS SA	18.40 ± 0.929	16.71 ± 2.075	91	22.28 ± 2.848	27.57 ± 2.182	124
FC SA	14.97 ± 1.351	15.78 ± 1.281	105	16.65 ± 3.058	19.86 ± 3.718	119
IMT	4.29 ± 0.373	4.25 ± 0.154	99	3.99 ± 0.185	4.08 ± 0.295	102
TDC	0.0069 ± 0.0006	0.0068 ± 0.0008	99	0.0085 ± 0.0011	0.0104 ± 0.0019	123
SDC	0.0205 ± 0.0020	0.0200 ± 0.0033	98	0.0072 ± 0.0007	0.0092 ± 0.0020	129

Values are mean ± SE; n = 9 per genotype at E16 and n = 6 per genotype at E19 from 3 litters/group; LT—labyrinthine trophoblast (mm^3^); MBS—maternal blood spaces (mm^3^); FC—fetal capillaries (mm^3^); MBS SA—maternal blood spaces surface area (cm^2^); FC SA—fetal capillaries surface area (cm^2^); IMT—interhemal membrane thickness of the labyrinthine zone (μm); TDC—theoretical diffusion capacity of the interhemal membrane (cm^2^.min^−1^.kPa^−1^); SDC—specific diffusion capacity (cm^2^.min^−1^.kPa^−1^.g^−1^); * *p* < 0.05.

**Table 4 genes-12-00639-t004:** Effects of deletion of selected imprinted genes on placental transport characteristics at E19 of mouse pregnancy relative to their wild-type littermate controls.

Gene Deletion	Theoretical Diffusion Capacity	Passive Permeability	Transporter Mediated Transport		Placental Size		Normal Role of Gene on Passive Permeability	Reference
			MeAIB	Glucose	Lz	Jz		
*Phlda2*	No Δ	↑ (inulin)	No Δ	No Δ	No Δ	↑	Restricts paracellular pore length, radius or number	This study
*Igf2*P0	↓	↓ (inulin, EDTA, mannitol)	No Δ	↑	↓	↓	Increases TDC by reducing interhemal membrane thickness and increasing surface area	[23,24]
*Igf2*	↓	↓ (inulin)	↓	No Δ	↓	↓	Increases TDC by reducing interhemal membrane thickness and increasing surface area	[24,45]
*H19*^Δ*13*^ (leading to *Igf2* LOI)	↑	↓ (inulin, mannitol)	↓	↓	↑	↑	Increases paracellular pore characteristics and reduces TDC through effects on surface area	[42]

Lz—labyrinthine zone, Jz—junctional zone, No Δ—no change, ↑—increase, ↓—decrease, TDC—theoretical diffusion capacity of the interhemal membrane, LOI – Loss of Imprinting.

## Data Availability

Not applicable.

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
