# Peer review of "Deletion of the Imprinted Phlda2 Gene Increases Placental Passive Permeability in the Mouse"

_genes, 2021, doi:10.3390/genes12050639_

Round 1
Reviewer 1 Report
Angiolini et al report on the consequence of loss-of-function of the imprinted Phlda2/Ipl gene on placental transport. As suggested by previous studies on this model, the authors report normal morphology of the labyrinthine zone at E16 and E19 after applying a detailed phenotyping protocol. They report a 4% reduction in fetal weight at E16 with no weight difference at E19 essentially consistent with Frank et al., 2002 “small (13%) but significant decrease in the mean fetal weight at E16.5” and Salas et al 2004 “We previously reported slight fetal growth retardation in the isolated Phlda2 knockout (Frank 2002) and this effect was also seen in the current data at 14.5 dpc, though not significantly at 16.5 dpc”. The authors find no evidence for changes in either active or facilitated transfer (14C‐methyl-D-glucose; 14C‐MeAIB) and some evidence of increased passive diffusion (14C-mannitol; 14C-inulin). The authors conclude “Our findings uncover a key role played by the imprinted Phlda2 gene in regulating placental passive permeability that may be important for determining fetal growth.”
- Terms such as “regulates”, “modulates” and “controlling” imply a direct and functional relationship between Phlda2 and passive permeability in the placenta. The authors report that loss of Phlda2 is associated with more 14C-mannitol and 14C-inulin in mutant fetuses relative to controls, for which there may be other explanations. Is there a technique to directly test the relationship between Phlda2 and passive permeability? If not, less definitive language should be used in the title and elsewhere.
- The authors suggest that changes in passive permeability relate to the fetal growth characteristics in this model. Placental growth plateaus before E19 when the authors report no difference in fetal weights. It is possible that the change in fetal growth dynamics relate to relatively lower demand for placental growth at this later stage of gestation. This possibility should be discussed in considerably more detail in the discussion.
- Fetal growth restriction is a clinical term used classify a significant deviation from the norm generally of >10%. The weight difference reported here is 4% which would not normally be described as fetal growth restriction.
- Salas et al 2004 previously reported a transient reduction in fetal growth in this model and Tunster et al 2016 reported a global reduction. These studies should be included in the introduction.
- Primary data for 14C-mannitol, 14C-inulin, 14C‐methyl-D-glucose and 14C‐MeAIB should be presented by genotype and not as ratios of Phlda2-/+ to Phlda2+/+.
- Mannitol and inulin are inert, uncharged and hydrophilic molecules used to measure passive diffusion. What molecules in this category are relevant to fetal growth?
- The description of where Phlda2 is expressed in the mouse placenta could be improved. For example “PHLDA2 protein is highly expressed in the villous cytotrophoblast of human placentae [10], an area analogous to the type II trophoblast cells in the murine placenta, where PHLDA2 expression is also highest [11]” Phlda2 is most highly expressed in the ectoplacental cone and yolk sac (Dunwoodie Int J Dev Biol 2002) and expression is relatively low in the syncitiotrophoblast of the labyrinth. “glycogen cells of the junctional zone of the mouse placenta, where expression is also very weak ” Phlda2 is not expressed in the glycogen cells of the junctional zone.
- “Within the junctional zone, Phlda2-/+ mutants have increased accumulation of glycogen and increased expression of several placental hormone genes [11,12] “ Rephrase as ““Within the junctional zone, Phlda2-/+ mutants have more glycogen stored [11,12] and the expression of several placental hormone genes are increased (reference is Tunster Dev Biol 2016)”
- “Maternal inheritance of Phlda2-/+ deletion led to placental overgrowth, as previously reported [11]” should be rephrased as “Maternal inheritance of Phlda2-/+ deletion on the C57BL6 background led to placental overgrowth and a transient reduction in fetal growth, as previously reported [11] (Salas et al 2004)”
- Was the data collected for the fully wild type samples concurrent to the study on Phlda2-/+ deletion?
- “These findings taken together are consistent with the suggestion that the overgrowth of Jz is controlled by signalling events taking place in Lz by unknown mechanisms [11].” The alternative explanation that Phlda2 limits the expansion of progenitors contributing to the Jz (Tunster MCB 2010) should be mentioned.
- “However, we cannot rule out other explanations such as increased partitioning of maternal resources to the placenta to sustain the expansion of the Jz layer” This explanation has been previously suggested and should be acknowledged (ref is Tunster Dev Biol 2016).
Reviewer 2 Report
Angiolini and coworkers report on physiological characterisations of Phlda2 mutant mice. They describe their findings from stereology studies in a mouse model deficient for the maternally expressed Phlda2 gene, as well as from placental transfer experiments. They could show for Phlda2 as the first imprinted gene that it regulates passive placental permeability, and concluded that it thereby has an impact on fetal growth.
In general, the paper is well written, documentation is satisfactory and conclusions convincingly drawn. Therefore I don´t have any comments about the methods and results, but what is missing is that the authors speculate on the relevance of their findings for the functional mechanisms of the 11p15 imprinted region(s). In fact, in the last paragraph of the discussion, they summarize the current knowledge on Igf2 and H19 and their role, but it would be interesting to learn how Phlda2 and these genes/gene products interact. Would aberrant imprinting at the H19/Igf2 DMR affect Phlda2 expression, is Phlda2 part of an imprinting network (Rtl2 has been mentioned on another chromosome)?
Can these data help to understand the biological link between disturbed imprinting in 11p15.5 and overgrowth/growth retardation phenotypes? Might the authors speculate on an interaction between different imprinted loci?
Reviewer 3 Report
In this study, the authors investigated the role of the Phlda2 gene in the mouse placenta. They employed the previously described Phlda2 knockout mouse line that is characterized by placentomegaly and fetal growth restriction. To investigate the basis of this phenotype, they first performed a number of stereological analyses of the placenta. Despite Phlda2 expression is higher in the labyrinthine zone than in the junctional zone of the placenta, they observed that the junctional zone of the placenta was enlarged in the mutant mice, while the size and structure of the labyrinthine zone were not affected. They then investigated the in vivo transfer of several solutes through the placenta and demonstrated that the passive diffusion of hydrophilic solutes was enhanced in the mutant mice, while other transport processes, such as active transport and facilitated diffusion, were not affected. The authors propose that the enhanced placental diffusion is responsible for the in utero catch-up growth they observed in the late fetal development of the mutant mice.
The study is well performed and adds further to a number of known mechanism by which imprinted genes control nutrient transfer through the placenta. However, the molecular mechanism by which Phlda2 controls placental transfer remains unclear. The authors may speculate a bit more on this issue in the Discussion. Furthermore, to help the reader in the last part of the Discussion, it is necessary to add a summary figure, which includes the regulatory functions of imprinted genes on placenta-fetus exchange of nutrients demonstrated so far.
Round 2
Reviewer 1 Report
The authors have addressed the comments made which improve the reading and wider interpretation of their research. I appreciate that the authors have now included the original data albeit as a supplemental figure. It is reassuring to see that the significance reported through expressing their data as ratios is maintained in the analysis of the original measures.